# Improved assessments of bulk milk microbiota composition via sample preparation and DNA extraction methods

**Zhengyao Xue**¤, **Maria L. Marco**◯*

Department of Food Science & Technology, University of California, Davis, California, United States of America

¤ Current address: Impossible Foods, Inc, Redwood City, California, United States of America
* mmarco@ucdavis.edu

**Data Availability Statement:** DNA sequences are deposited in the Qiita database under study ID 11351 and 12369, and in the European Nucleotide Archive (ENA) under accession number ERP104377 and ERP116294.

## Abstract

Although bacterial detection by 16S rRNA gene amplicon DNA sequencing is a widely-applied technique, standardized methods for sample preparation and DNA extraction are needed to ensure accuracy, reproducibility, and scalability for automation. To develop these methods for bovine bulk milk, we assembled and tested a bacterial cell mock community (BCMC) containing bacterial species commonly found in milk. The following protocol variations were examined:: BCMC enumeration (colony enumeration or microscopy), sample volume (200 µl to 30 ml), sample storage condition (frozen in PBS or 25% glycerol or exposure to freeze-thaw cycles), cell lysis method (bead-beating, vortex, enzymatic), and DNA extraction procedure (MagMAX Total, MagMAX CORE, and MagMAX Ultra 2.0, with and without either Proteinase K or RNase A). Cell enumeration by microscopy was more accurate for quantification of the BCMC contents. We found that least 10 mL ($\geq 10^4$ cells in high quality milk) is needed for reproducible bacterial detection by 16S rRNA gene amplicon DNA sequencing, whereas variations in storage conditions caused minor differences in the BCMC. For DNA extraction and purification, a mild lysis step (bead-beating for 10 s at 4 m/s or vortexing at 1800 rpm for 10 s) paired with the MagMAX Total kit and Proteinase K digestion provided the most accurate representation of the BCMC. Cell lysis procedures conferred the greatest changes to milk microbiota composition and these effects were confirmed to provide similar results for commercial milk samples. Overall, our systematic approach with the BCMC is broadly applicable to other milk, food, and environmental samples therefore recommended for improving accuracy of culture-independent, DNA sequence-based analyses of microbial composition in different habitats.

## Introduction

Targeted 16S rRNA gene sequence analysis by PCR amplicon, high throughput DNA sequencing is now the most widely used technique to study environmental, food, animal, and human microbiota due to its relatively low cost, ease of use, and sensitivity for detection of low numbers of different bacterial taxa. However, the sample preparation and DNA sequencing

**Funding:** The study was funded by the following: California Dairy Research Foundation grant "Rapid Methods for Microbial Detection and Quality Assessment of Milk". The funders had no role in study design, data collection and analysis, decision to publish, or preparation of the manuscript.

**Competing interests:** The authors have declared that no competing interests exist.

workflow still involves multiple steps which are vulnerable to methodological biases and contamination issues [1, 2]. For example, primer choice [3, 4], DNA polymerase [5, 6], PCR cycle number [6, 7], library preparation [5], DNA sequencing [8–10], and bioinformatics methods [11–13] can each introduce variation, affecting the interpretation of bacterial community composition regardless of sample type. In addition, careful planning is needed to include the proper negative controls to detect contamination [14, 15].

Sample type (for example soil, feces, water, and foods) also requires methodological consideration for sample quantity, storage, cell lysis, and DNA extraction and purification protocols. Matrix properties (e.g. PCR inhibitors) and bacterial species and numbers are extremely variable between microbial habitats. Challenges to the examination of the bovine milk microbiota are notable because even high-quality bulk milk, containing low numbers of total bacteria ($10^3$ to $10^4$ cells/ml) still contains a diverse microbiota with many bacterial species [16–18]. The nuance to this issue is expanded even further when evaluating freshly expelled milk from individual healthy and mastitic cows which appear to have DNA recalcitrant to PCR amplification [19, 20]. These characteristics and the presence of high levels of proteins and fats in milk show the need for optimized bacterial DNA extraction protocols. Ideally, methods should also be amenable to automation and appropriate for analyzing massive numbers of milk samples in parallel.

We previously compared different DNA sequencing and analysis methods for the identification of bacteria using purified gDNA and PCR amplicons from a bacterial cell mock community (BCMC) comprised of nine bacterial species commonly found in milk [9]. In this study, our goal was to use the BCMC to develop a milk sample processing and DNA extraction workflow that is suitable to automation (**Fig 1**).

## Materials and methods

### Bacterial strains and growth conditions

Bacterial strains from species commonly found in milk and other dairy products were selected for a BCMC (**Table 1**). The strains were grown as previously described [9] in the following culture media: LB (Lennox broth; Thermo Fisher Scientific, Waltham, WA, USA) for *Bacillus subtilis* S44, *Pseudomonas fluorescens* A506, and *Escherichia coli* ATCC 700728; brain heart infusion broth (Thermo Fisher Scientific, Waltham, WA, USA) for *Enterococcus faecalis* ATCC 29212 and *Streptococcus agalactiae* ATCC 27956; tryptic soy broth (Becton Dickinson, Franklin Lakes, NJ, USA) for *Staphylococcus aureus* ATCC 29740 and *Corynebacterium bovis* ATCC 7715; M17 broth (Becton Dickinson, Franklin Lakes, NJ, USA) with 0.5% w/v glucose for *Lactococcus lactis* IL1403; and reinforced clostridial broth (Becton Dickinson, Franklin Lakes, NJ, USA) for *Clostridium tyrobutyricum* ATCC 25755.

### Preparation of the BCMCs

The bacterial cell mock community (BCMC) was prepared twice. For BCMC1, the nine bacterial strains were grown to early stationary phase and 20 μL aliquots of each culture were collected and combined into multiple, multi-species pools. At the time of BCMC1 preparation, cell numbers were estimated by plating serial dilutions of each strain onto the appropriate laboratory medium for incubation and enumeration of colony forming units (CFU) (**Table 1**). Freshly prepared BCMC1 pools were either used directly for DNA extraction or stored at -80˚C. For BCMC2, the nine strains were grown to early stationary phase and cell numbers were directly enumerated by microscopy (Standard 20, ZEISS Microscopy, Jena, Germany) using a hemocytometer (Hausser Scientific, Horsham, PA, USA). The strains were then combined in equal numbers (**Table 1**) based on the direct cell counts and the BCMC aliquots were

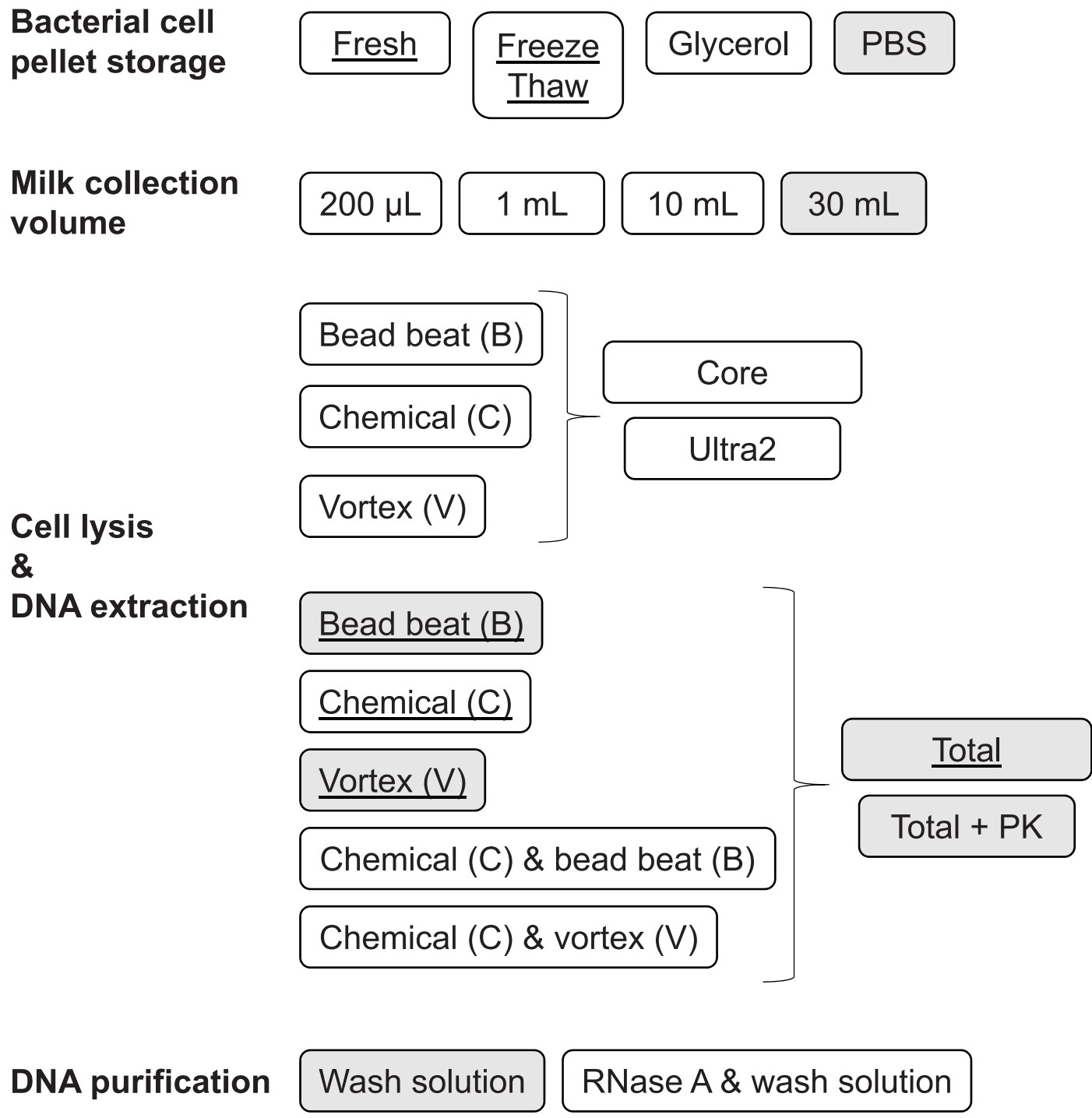

**Fig 1. Schematic diagram of the experimental design.** The BCMC was tested for the effects of variations in sample preparation, cell lysis, DNA purification and collection methods. Cells were lysed using a bead beater (B), vortex (V), or incubated for chemical lysis (C). Wash solution denotes the solutions included in DNA extraction kits used for purification. Wash step(s) were performed following manufacturer's protocol. Underlined text indicates experiments performed with both BCMC1 and BCMC2. For all other experiments, BCMC2 was used. Filled boxes are methods used to test raw bulk milk samples.

**Table 1. Bacterial strains and expected relative abundances in the BCMC.**

| Strain | 16S rRNA gene copy number | Expected percentage (%) [a] | | References |
|---|---|---|---|---|
| | | BCMC1 | BCMC2 | |
| *Bacillus subtilis* S44 | 10 | 7.40 | 18.16 | [21] |
| *Clostridium tyrobutyricum* ATCC 25755 | 1 | 3.69 | 12.28 | [22] |
| *Corynebacterium bovis* ATCC 7715 | 1 | 0.31 | 1.67 | [23] |
| *Enterococcus faecalis* ATCC 29212 | 4 | 13.38 | 7.21 | [24] |
| *Escherichia coli* ATCC 700728 | 7 | 12.26 | 13.64 | [25] |
| *Lactococcus lactis* IL1403 | 6 | 24.90 | 10.81 | [26] |
| *Pseudomonas fluorescens* A506 | 6 | 24.33 | 11.59 | [27] |
| *Staphylococcus aureus* ATCC 29740 | 5 | 5.98 | 10.31 | [28] |
| *Streptococcus agalactiae* ATCC 27956 | 7 | 7.74 | 14.33 | [29] |

[a] BCMC1 and BCMC2 were prepared using CFU/ml and direct microscopy for cell enumeration, respectively. Expected percentages are estimated based on estimated cell numbers in the BCMC and 16S rRNA gene copy numbers for each species.

washed twice with Phosphate Buffered Saline (PBS) (pH 7.4). Freshly prepared BCMC2 pools were either used for directly for DNA extraction or stored at -80˚C.

## Sample storage condition comparisons

To avoid the storage of large milk sample volumes, bacteria can be collected from milk by centrifugation prior to preservation at -20˚C and subsequent analysis [16, 30, 31]. To investigate the effect freezing on BCMC composition, DNA was extracted from BCMC1 either immediately after preparation (Fresh) or after five freeze-thawed cycles between -20˚C and 4˚C (Freeze Thaw) (**Fig 1**). BCMC2 was stored at -20˚C in either PBS or PBS with 25% v/v glycerol for 7 days prior to DNA extraction.

## Milk collection volume comparisons

To closely simulate the bacterial cell collection process from bulk milk via centrifugation, freshly prepared BCMC2 were mixed with ultra-high temperature (UHT) pasteurized milk (UHT 2% reduced-fat milk, Gossner Foods, Inc., Logan, UT) to reach a quantity of $6.32 \times 10^3$ cells/mL, an amount similar to the numbers of bacteria in high quality raw, bulk milk [32] (**Fig 1**). UHT milk was used in order to minimize background microbial contamination. Aliquots of 200 μL, 1 mL, 10 mL, and 30 mL were then centrifuged at 13,000 g for 5 min at 4˚C to collect the inoculated BCMC2, corresponding to $1.26 \times 10^3$ cells, $6.32 \times 10^3$ cells, $6.32 \times 10^4$ cells, and $1.90 \times 10^5$ cells, respectively. Next, the cell pellets were washed twice with PBS and stored at -20˚C prior to DNA extraction.

## Cell lysis and genomic DNA extraction

Comparisons of milk sample volumes on estimates of bacterial composition were performed using genomic DNA from BCMC2. For assessing the effects of storage conditions, genomic DNA from either BCMC2 or BCMC1 and BCMC2 were used. For both sets of comparisons, BCMC genomic DNA was extracted using the MagMAX Total nucleic acid isolation kit (Thermo Fisher Scientific, Vilnius, Lithuania) following the manufacturer's protocol. Mechanical lysis was used as performed previously [9, 30] by two runs in a FastPrep-24 instrument (MP Biomedicals, Burlingame, CA, USA) at a setting of 6.5 m/sec for 1 min with a 1 min intervening interval on ice.

Comparisons of DNA extraction and purification steps were performed with BCMC2 and genomic DNA was extracted using three magnetic-bead based, DNA extraction kits [Mag-MAX Total Nucleic Acid Isolation kit (Total), MagMAX CORE Nucleic Acid Purification kit (Core), and MagMAX DNA Multi-Sample Ultra 2.0 Kit (Ultra2)] (Fig 1). Those kits were selected because they are compatible with the automated KingFisher Flex system (Thermo Fisher Scientific, Waltham, WA, USA). The three kits use silica-coated magnetic beads for binding DNA, guanidinium thiocyanate and 80% ethanol for DNA purification, and water for DNA elution. The Total kit included prefilled tubes with 600 mg 0.1mm zirconia beads. For the Core and Ultra2 kits, 500 mg of 0.1 mm zirconia beads (Thermo Fisher Scientific, Waltham, WA, USA) were added separately. For each of the kit comparisons, BCMC cells were lysed using the B1, V1, and C1 methods (Table 2). After collecting the BCMC lysates, gDNA purification was completed on the KingFisher Flex system following the manufacturer's protocol.

Other comparisons of BCMC cell lysis methods were performed with the MagMAX Total Nucleic Acid Isolation kit (Total) without automation. These methods were classified as mild or rigorous according to the lysis method used (Table 2). Rigorous conditions were tested on BCMC1 and included the B1, B2, B3, B4, V1, and V2 methods (Table 2). Mild cell lysis methods were tested at a later time with BCMC2 and included the B5, V3, and C2 methods and C2 +B5 and C2+V3 combinations (Table 2). After the lysis step for each of those conditions, a fraction of the lysate was further treated with either 200 μg/mL Proteinase K (PK, Thermo Fisher Scientific, Waltham, WA, USA) or 50 μg/mL RNase A (Thermo Fisher Scientific, Waltham, WA, USA) (Fig 1). For PK, the lysate was incubated at 70°C for 20 min with intermittent tube inversion. For RNase A, the lysate was incubated at 37°C for 15 min with intermittent tube inversion. Genomic DNA was then purified following the Total protocol.

## DNA extraction from commercial milk samples

Two raw (unpasteurized) milk from the storage silos and one HTST pasteurized milk samples were collected by a commercial dairy processor (Hilmar Cheese Company, Hilmar, CA) on 5/23/2018. Bacteria in the milk were collected as previously described [16, 30, 31]. Briefly, each milk sample was split into three 25 mL aliquots, centrifuged at 13,000 g for 5 min at 4°C, and then the cell pellets were stored at -20°C until. For DNA extraction, the cell pellets were subjected to either method B1 (Table 2) followed by DNA purification with the Total protocol, or

**Table 2. Cell lysis and DNA extraction methods tested.**

| Abbreviation | Cell lysis method | DNA extraction kit [a] |
|---|---|---|
| B1 | Bead-beating at 6.5 m/sec × 1 min × 2 | Total, Core, and Ultra2 |
| B2 | Bead-beating at 6.5 m/sec × 1 min | Total |
| B3 | Bead-beating at 3.5 m/sec × 1 min × 2 | Total |
| B4 | Bead-beating at 3.5 m/sec × 1 min | Total |
| V1 | Vortexing at 1800 rpm × 15 min | Total, Core, and Ultra2 |
| V2 | Vortexing at 1800 rpm × 10 min | Total |
| C1 | Chemical lysis at 80°C × 20 min + 37°C × 60 min | Total, Core, and Ultra2 |
| B5 | Bead-beating at 4 m/sec × 10 sec | Total, Total + PK |
| C2 + B5 | Chemical lysis at 37°C × 60 min + Bead-beating at 4 m/sec × 10 sec | Total, Total + PK |
| V3 | Vortexing at 1800 rpm × 30 sec | Total, Total + PK |
| C2 + V3 | Chemical lysis at 37°C × 60 min + Vortexing at 1800 rpm × 30 sec | Total, Total + PK |

[a] Total stands for MagMAX Total nucleic acid kit; Core stands for MagMAX Core DNA kit; Ultra2 stands for MagMAX Ultra 2.0 DNA kit. PK = proteinase K.

B5 or V4 (**Table 2**) followed by DNA purification with the Total protocol and Proteinase K treatment.

## 16S rRNA gene sequencing and analysis

Barcoded PCR of the 16S rRNA gene V4 region was performed as previously described [9] using the F515 (GTGCCAGCMGCCGCGGTAA) and R806 (GGACTACHVGGGTWTCTAAT) primers, targeting the 16S rRNA gene V4 region, with 8-bp random barcoded sequences on the 59 end of the forward primer [33]. and ExTaq DNA polymerase (TaKaRa, Otsu, Japan). Pooled and purified 16S V4 products were sequenced with the Ion Torrent PGM sequencer using the HiQ view 400 bp sequencing kit and a 318 v2 chip (Life Technologies, Carlsbad, CA).

Ion Torrent output BAM files were converted to FASTQ files using BEDTools [34]. Reads that were shorter than 200 bases were discarded. The remaining reads were analyzed using QIIME 1.9.1 [35] and QIIME 2 version 2018.4 [36] with previously described parameters [16, 30, 31]. QIIME 2 generated feature tables, feature sequences, rooted phylogenetic tree, and sample information were imported in R 3.4.2 and analyzed as previously described [16, 30, 31]. Briefly, sequence files were demultiplexed with the *demux* plugin. Feature table and chimera removal were performed using the DADA2 method. For alpha and beta diversity analysis, DNA sequences were rarefied to 3,000 reads per sample. Weighted differences of taxa proportions compared to expected were calculated using the following formula: weighted difference = (observed%—expected%) / expected%. For taxonomy assignment, a custom classifier was trained based on the truncated sequence reads (231 bases) against the Greengenes database version 13.8 [37] and accordance with our prior study showing the utility of this database for accurate identification of bacteria in milk [9].

## Accession numbers

DNA sequences after quality filtering and trimming were deposited in the Qiita database [38] under study ID 11351 and 12369, and in the European Nucleotide Archive (ENA) under accession number ERP104377 and ERP116294.

## Results

### Effect of BCMC preparation method on estimates of community membership

Two BCMCs were prepared using the same strains but different methods to estimate cell numbers at the time of BCMC preparation. Strain enumeration for BCMC1 was based on CFUs, whereas BCMC2 relied on direct counting by microscopy. Cell number estimates and the 16S rRNA gene copy numbers were then used to set the "expected" proportions of each member of the mock community (**Table 1**).

The 16S rRNA V4 region was sequenced and analyzed for each BCMC (n = 15 for BCMC1 and n = 33 for BCMC2) to result in "observed" proportions. For BCMC1, the observed proportions of *Escherichia* (9.28% ± 0.55%) and *Pseudomonas* (17.80% ± 3.45%) were significantly lower and the proportions of *Bacillus* (16.55% ± 2.91%) and *Corynebacterium* (3.19% ± 1.13%) were significantly higher than the expected values (**Fig 2**). This was improved for BCMC2 and only the proportions of *Lactococcus* (5.22% ± 6.48% increase) were significantly changed relative to the expected proportions (**Fig 2**). For BCMC2, although the levels of *Bacillus*, *Clostridium*, *Escherichia*, and *Pseudomonas* in BCMC2 were somewhat higher and *Streptococcus* lower than expected, those differences did not reach significance (p = 0.05, DESeq2 adjusted p < 0.1 and log$_2$ fold change > 1.5). Therefore, we concluded that the use of direct cell counting resulted in a better representation of cell numbers contained in the BCMC.

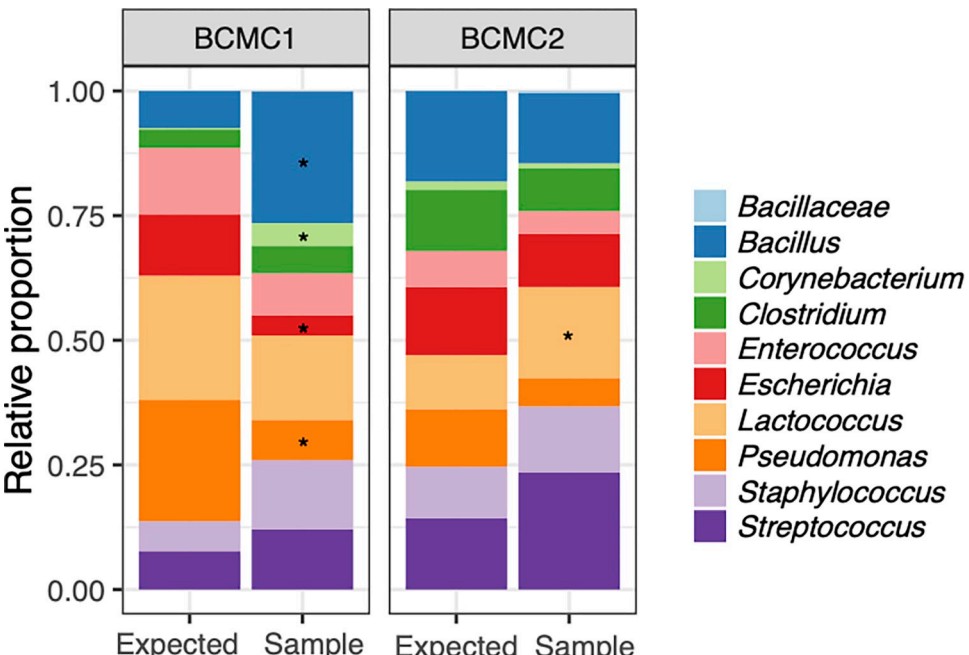

**Fig 2. Comparison of BCMC cell enumeration methods.** The expected values are based on CFU enumeration for BCMC1 and direct counts by microscopy for BCMC2. The observed results represent the average of 15 replicates of BCMC1 and 33 replicates of BCMC2 according to 16S rRNA gene amplicon DNA sequencing. BCMC replicates were processed the same by MagMAX Total kit and lysed with bead beating at 6.5 m/sec for 1 min twice with 1 min interval on ice prior to PCR. Significant changes (DESeq2 adjusted $p < 0.1$ and $\log_2$ fold change $> 1.5$) compared to the expected values are indicated by the presence of asterisks.

## Milk sample volume and cell number effects on estimates of bacterial composition

DNA extractions from larger collection volumes ($\geq 10$ mL) and hence higher numbers of total bacteria ($\geq 6.32 \times 10^4$ cells) led to a lower alpha diversity more similar to the expected values (**Fig 2A**), reduced intra-sample variability (**Fig 3B**), as well as improved the reproducibility of taxonomic distributions between BCMC replicates (**Fig 3C**).

Larger milk samples ($\geq 10$ mL) also contained fewer unexpected taxa assignments, or in other words, bacteria that were not a part of the mock community ("Other" in **Fig 3C**). The larger milk samples ($\geq 10$ mL) were detected as containing 1.4% to 3.8% unexpected bacterial taxa according to 16S rRNA gene sequencing; whereas 12% to 22% unexpected taxa were observed when $\leq 1$ mL milk ($6.32 \times 10^3$ cells) was used. Notably, the majority of those unexpected bacteria (**Fig 3**) were identified as either *Micrococcus* (3.11% ± 3.41%) or *Tepidimonas* (8.67% ± 8.75%). These taxa likely originated from UHT milk because both genera were prevalent in UHT milk controls tested without the addition of mock community cells (*Micrococcus*: 2.4% ± 4.9% and *Tepidimonas*: 10.8% ± 6.3%). Therefore, we estimated that for the cell lysis and DNA extraction methods used here, at least $6.30 \times 10^4$ cells are needed, a number expected to be reached with a sample volume of 10 mL good quality, raw bulk milk.

## Storage method effects on mock community detection

BCMC bacterial proportions were not affected by freezing the cells at -20°C in either PBS or in water with 25% v/v glycerol (**Fig 4A**). There were also no changes in estimates of BCMC composition when prepared fresh or exposed to five freeze-thaw cycles (**Fig 4B**).

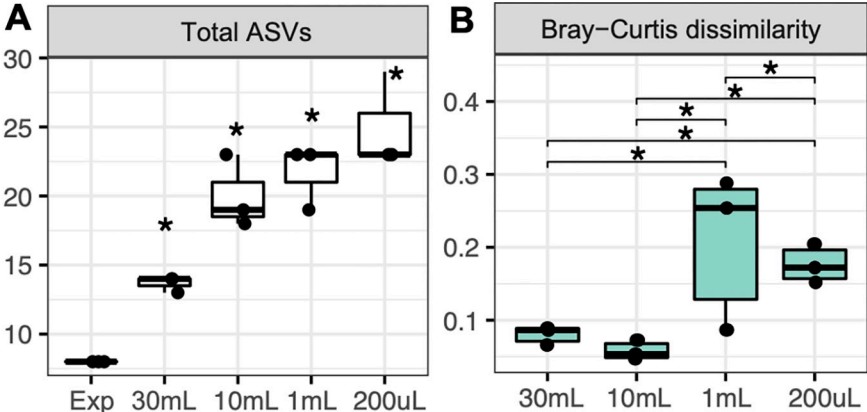

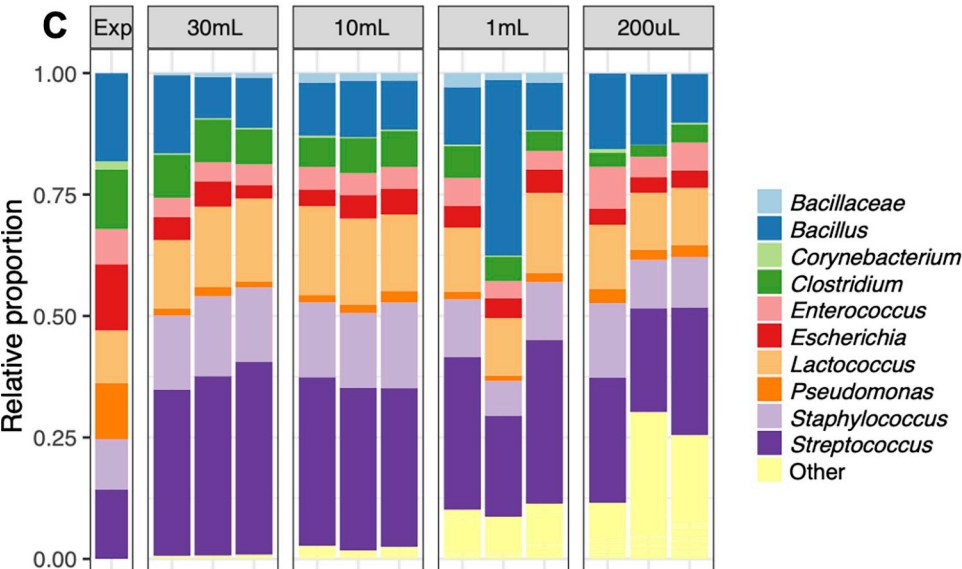

**Fig 3. Microbial diversity and composition of BCMC taxa detected upon recovery from different volumes of milk.**
BCMC2 was inoculated into UHT milk and sampled at the following volumes (cell numbers): 200 μL ($1.26 \times 10^3$ cells),
1 mL ($6.32 \times 10^3$ cells), 10 mL ($6.32 \times 10^4$ cells), and 30 mL ($1.90 \times 10^5$ cells). Expected percentages are estimated based
on estimated cell numbers in the BCMC and 16S rRNA gene copy numbers for each species. The **(A)** total number of
ASVs, **(B)** Bray-Curtis dissimilarity of intra-sample variation, and **(C)** expected taxa (9 bacterial species) are labeled
with the corresponding taxonomic level from DNA sequencing results. Unexpected taxa were labeled as "Other". Each
bar represents a single replicate of BCMC2. Significant differences (Kruskal-Wallis with Dunn test, $p < 0.05$) from the
expected value **(A)** and between sample groups **(B)** are indicated by the presence of asterisks.

## Comparison of cell lysis and DNA purification methods

To develop a streamlined method for bacterial DNA purification from milk, we examined the
BCMC with different combinations of cell lysis methods (bead-beating, vortex and chemical
lysis) and magnetic-bead based DNA purification kits (MagMAX Total, Core and Ultra2 kits)
amenable for use on a widely-used, stand-alone automation instrument (**Table 2**).

Using kit-recommended, rigorous lysis methods (B1, V1, and C1) (**Table 2**), the MagMAX
Total kit (Total) resulted in significantly lower Bray-Curtis dissimilarities and UniFrac dis-
tances relative to the expected values than the Core and Ultra2 kits (**Fig 5**). This finding was

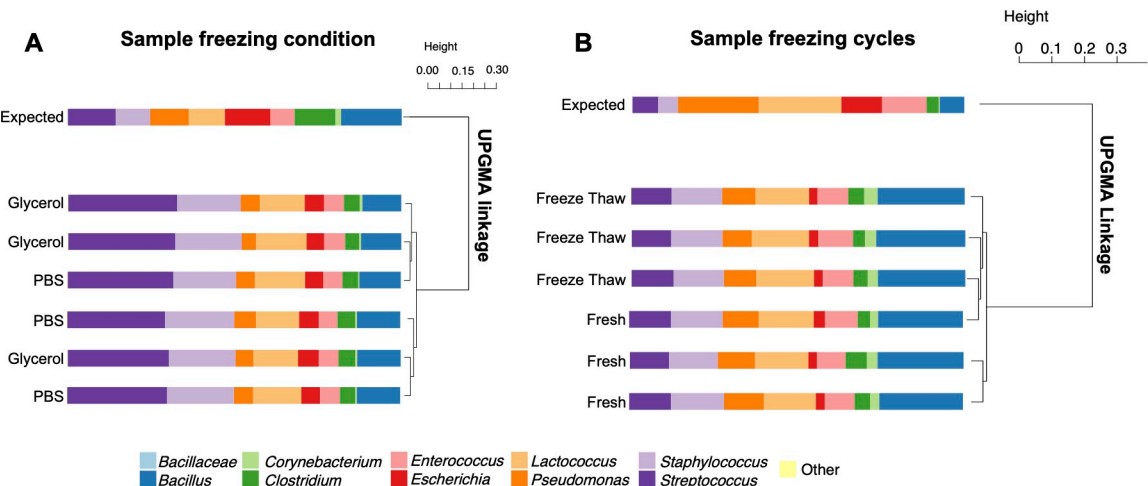

**Fig 4. Relative proportions of taxa and UPGMA hierarchical clustering of BCMC exposed to different storage conditions.**
UPGMA hierarchical clustering was based on Bray-Curtis dissimilarity matrix for (A) BCMC2 frozen with PBS or 25% v/v glycerol and (B) BCMC1 that were prepared fresh or exposed to five freeze-thaw cycles. Expected taxa (9 bacterial species) are labeled with the corresponding taxonomic level from 16S rRNA gene amplicon DNA sequencing results. Each bar represents a single replicate.

consistent between the three different cell lysis methods (B1, V1, and C1) applied, suggesting that the Total kit was better than the other two purification methods, irrespective of which rigorous lysis method was used (**Fig 5**). Although the C1 lysis method combined with the Ultra2 kit resulted in comparable Bray-Curtis dissimilarities and UniFrac distances as the Total kit with B1 lysis method (**Fig 5**), the combination of the C1 and Ultra2 methods also led to large increases in the estimates of *Lactococcus* relative abundance (3.12 ± 0.62 fold of weighted difference from the expected values) (**Fig 6**).

Rigorous cell lysis methods such as those represented by B1, V1, and C1 are known to introduce bias in bacterial community assessments [39, 40]. Therefore, we also tested other, milder approaches in combination with the MagMAX Total kit for DNA purification. Cell lysis using either B2 (beat-beating at either 1 min at 6.5 m/sec), B3 (1 min at 3.5 m/sec twice with 1 min interval on ice), or V2 (1 min at 3.5 m/sec), or B4, (vortexing for 10 min at 1800 rpm) (**Table 2**) did not affect the outcomes of BCMC assessments compared to the more rigorous, kit-recommended method B1 (bead-beating twice for 1 min at 6.5 m/sec with 1 min interval on ice) (**Fig 7**). Like found for B1, each of those milder lysis variations resulted in significantly reduced proportions of *Escherichia* and *Pseudomonas* and increased proportions of *Bacilliaceae*/*Bacillus*, *Corynebacterium*, and *Staphylococcus* relative to the expected values (**Fig 7**).

Because *Escherichia* and *Pseudomonas* are Gram-negative bacterial taxa, these members of the BCMC may be more easily lysed and vulnerable to DNA shearing than Gram-positive bacteria. Therefore, we tested even milder cell disruption methods, namely B5 (bead-beating for 10 sec at 4 m/sec) and V3 (vortex at 1800 rpm for 30 sec) with and without lysozyme treatment (C2) (**Table 2**). Application of these methods greatly improved our capacity to identify bacteria in the BCMC at their known proportions. Only between one to five BCMC taxa were detected in significantly altered levels by 16S rRNA gene DNA sequencing relative to expected values based on direct enumeration by microscopy (**Fig 6**). By comparison, between four to eight strains were found in significantly altered proportions when more rigorous lysis methods were used (**Fig 6**).

We also tested the addition of RNase A (50 µg/mL) and Proteinase K (PK, 200 µg/mL) after cell lysis to digest any remaining RNA contaminants and milk protein, respectively.

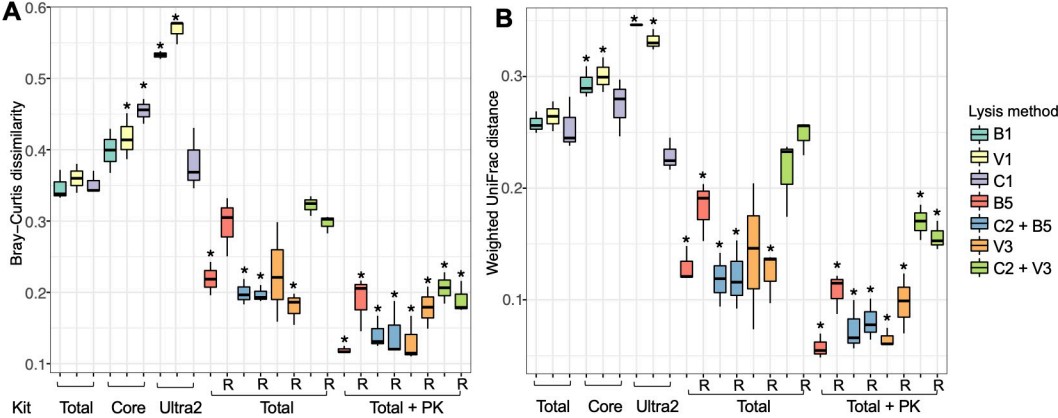

**Fig 5. Distances between the expected values and mock community samples. (A)** Bray-Curtis dissimilarity and **(B)** weighted UniFrac distances compared to expected values. Significant differences (Kruskal-Wallis with Dunn test, p < 0.05) from the standard DNA extraction method (Total kit with B1 lysis method) are indicated by the presence of asterisks. The following lysis methods were applied: B1 (bead-beating twice for 1 min at 6.5 m/sec with 1 min interval on ice), V1 (vortex at 1800 rpm for 15 min), C1 (incubation at 80˚C for 20 min and 37˚C for 60 min), B5 (bead-beating for 10 sec at 4 m/sec; C2 (incubation at 37˚C for 60 min), and V3 (vortex at 1800 rpm for 30 sec). Total (MagMAX Total nucleic acid kit), Core (MagMAX Core DNA kit), Ultra2 (MagMAX Ultra 2.0 DNA kit); and PK (proteinase K). Abbreviations for the cell lysis and DNA purifications methods are also provided in Table 2. The letter R on the x-axis denotes RNase A treated samples.

Surprisingly, RNase A treatment resulted in significantly higher proportions of unexpected taxa (7.24% ± 6.16% of total bacteria detected) than untreated controls (0.41% ± 0.21%, Mann-Whitney p < 0.05). The unexpected taxa in RNase A treated samples were predominantly *Ruminococcus* (0.81% ± 0.68%), *Roseburia* (0.78% ± 0.65%), and *Faecalibacterium* (0.73% ± 0.67%). By comparison, Proteinase K combined with mild BCMC cell lysis led to more accurate taxonomic proportions (B5 + PK: 0.33% ± 0.18% unexpected taxa of total bacteria detected, V3 + PK: 0.40% ± 0.17%) and lower Bray-Curtis dissimilarities and UniFrac distances from the expected BCMC beta diversities (**Fig 5**). Importantly, the taxonomic proportions were very similar to those predicted for the BCMC. Only the proportions of

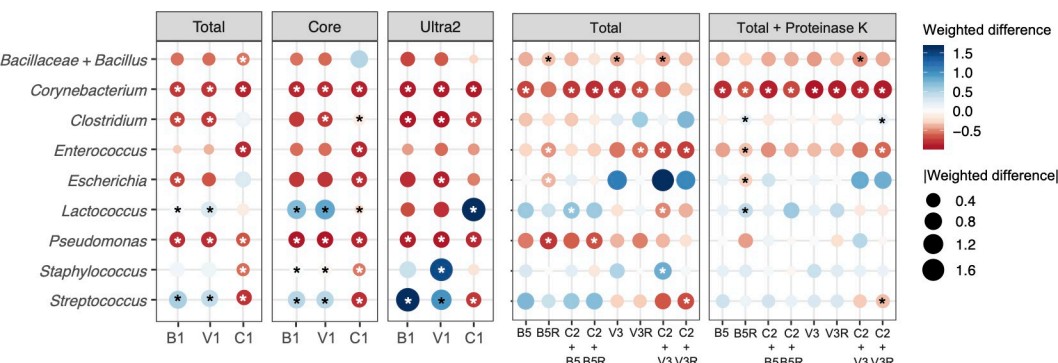

**Fig 6. Effects of cell lysis methods on BCMC proportions.** The weighted difference was calculated using the following formula: (observed%—expected%) / expected%. Each circle is the average value from three replicates of BCMC2. Different colors indicate the value of weighted differences and dot sizes indicate the absolute values of weighted differences. Significant changes (DESeq2 adjusted p < 0.1 and $\log_2$ fold change > 1.5) compared to the expected values are indicated by the presence of asterisks. Abbreviations are used as follows (see also Table 2): B1 (bead-beating twice for 1 min at 6.5 m/sec with 1 min interval on ice), V1 (vortex at 1800 rpm for 15 min), C1 incubation at 80˚C for 20 min and 37˚C for 60 min); B5 bead-beating for 10 sec at 4 m/sec), C2 stands for incubation at 37˚C for 60 min), and V3 (vortex at 1800 rpm for 30 sec). The letter "R" after lysing methods denotes RNase A treatment.

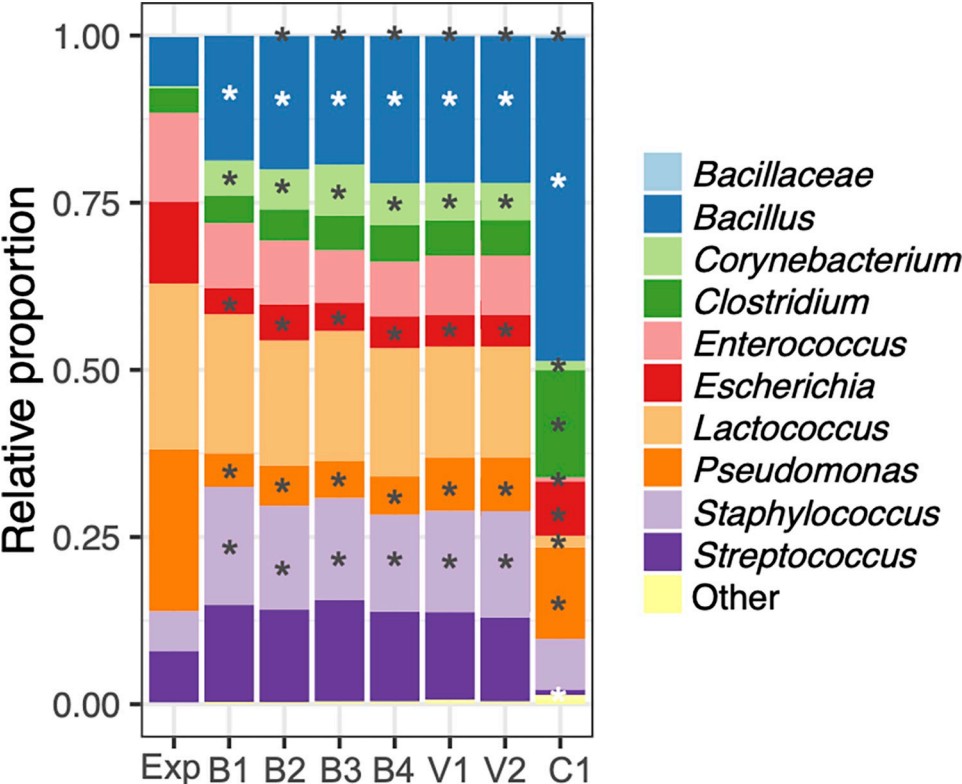

**Fig 7. Effects of B1, B2, B3, B4, V1, and V2 cell lysis methods on BCMC1 taxonomy.** DNA extractions were performed on BCMC1 using the MagMAX Total nucleic acid kit. Each bar represents the average value of three replicates. Significant differences (DESeq2 adjusted p < 0.1 and $\log_2$ fold change > 1.5) from the expected value are indicated by the presence of asterisks. Exp stands for expected, B1 (bead beat twice for 1 min at 6.5 m/sec with 1 min interval on ice), B2 (bead beat for 1 min at 6.5 m/sec), B3 (bead beat twice for 1 min at 3.5 m/sec with 1 min interval on ice), B4 (bead beat at 3.5 m/sec for 60 sec), V1 (vortex at 1800 rpm for 15 min), V2 (vortex at 1800 rpm for 10 min), and C1 (incubation at 80˚C for 20 min and 37˚C for 60 min).

*Corynebacterium* were significantly altered, and slightly reduced proportions were found compared to expected values (B5 + PK: -0.72 ± 0.01 weighted difference, V3 + PK: -0.79 ± 0.05 weighted difference) (**Fig 6**).

## Application of cell lysis and DNA extraction methods for assessing raw and pasteurized milk microbiota

Commercially prepared milk samples collected at different times during a production day before and after HTST pasteurization were examined by 16S rRNA gene amplicon DNA sequencing using the cell lysis and DNA purification methods which resulted in the most accurate BCMC proportions. Specifically, we tested B5 (bead-beating for 10 s at 4 m/s) and V3 (vortexing at 1800 rpm for 30 s) with PK treatment and the MagMAX Total kit for DNA purification. For comparison to standard cell lysis conditions, the more rigorous protocol B1 (Total/B1) was also used.

Estimates of bacterial abundance obtained for the raw and HTST milk samples showed that the proportions of bacterial taxa were impacted by the cell lysis method. *Bacillus* (*Bacillaceae*), *Lactococcus*, and *Staphylococcus* proportions were lower and *Escherichia*, *Pseudomonas* and *Streptococcus* were higher in milk extracted with B5 or V3 compared to the B1 lysis method (**Fig 8A**). Remarkably, the lysis method affected assessments of bacterial composition in the

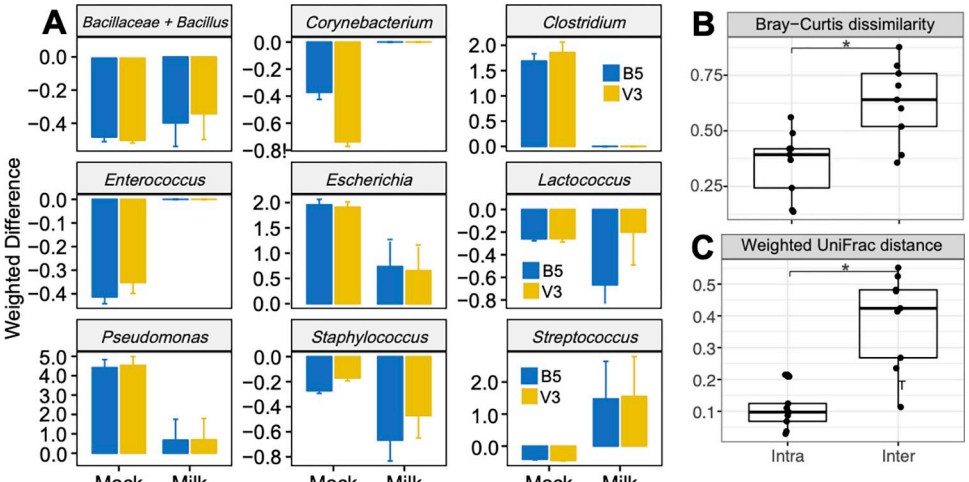

**Fig 8. Beta diversity and weighted differences of commercial milk samples and the BCMC. (A)** weighted differences of mock community samples compared to the Total/B1 method, **(B)** Bray-Curtis dissimilarity, and **(C)** weighted UniFrac distances between milk samples are shown. Weighted differences for each organism are calculated using the following formula: (B5% or V3%—B1%) / B1%. Significant differences (B and C, Kruskal-Wallis with Dunn test, p < 0.05) are indicated by the presence of asterisks. Abbreviations are used as follows (see also **Table 2**): B1 (bead-beating twice for 1 min at 6.5 m/sec with 1 min interval on ice using the MagMAX Total kit), B5 (bead-beating for 10 sec at 4 m/sec), and (vortex at 1800 rpm for 30 sec). B5 and V3 samples were extracted using the MagMAX Total kit with PK digestion.

BCMC and commercial milk samples in very similar ways. Only *Streptococcus* was oppositely affected by lysis method (**Fig 8A**). Levels of *Streptococcus* were slightly decreased in the BCMC (weighted difference of -0.41 and -0.44 (for B5 and V3, respectively, compared to B1); whereas proportions of this genus increased in the commercial milk samples (weighted difference of 1.47 and 1.54 for B5 and V3, respectively, compared to B1) (**Fig 8A**). Thus, taken together, these findings confirmed the value of using mock communities for method testing.

Despite the impact of cell lysis method on bacterial detection and quantification in milk, variation between different milk samples was significantly greater than intra-sample variation introduced by methodological changes (**Fig 8B and 8C**). These results show that although cell lysis methods may introduce biases into 16S rRNA gene amplicon DNA sequencing results, the methods used is not likely to impair comparative, microbiota studies between different milk samples.

## Discussion

The identification and tracking of bacterial populations by 16S rRNA gene sequence surveys offers significant opportunities to better understand and monitor those microorganisms in foods and other complex microbial habitats. However, the methods used are still susceptible to biases introduced at nearly every step from sample collection to data analysis and interpretation. Herein, we evaluated several parameters relevant to the identification of bacteria in bulk milk. Among these parameters, we determined that at least 10 mL milk (estimated to contain approximately $10^3$ cells/mL) combined with mild cell lysis methods, Proteinase K treatment and magnetic bead-based DNA purification (Total kit) provides the most accurate bacterial identification in a manner that is also suitable for automation. We also found that the use of a mock community, the identification of critical points for analysis, and the validation of proposed protocols with environmental (milk) samples are useful ways to evaluate these methodological considerations. This approach can be applied for investigating other microbial habitats

containing diverse and low numbers of bacteria. Combining these steps with development of validated downstream DNA sequencing and bioinformatics methods, as we showed previously [9], can provide a complete pipeline for bacterial community analysis within specific environments and sample matrices (e.g. foods, soils, etc). Additional considerations for the different variables in bulk milk sample preparation are provided below.

## Milk sample volume

BCMC alpha and beta diversity increased significantly when testing low quantities of BCMC cells (consistent with bacterial numbers expected to be present in milk ($< 10$ mL)). This variation in microbial content is indicative of background contamination from the UHT milk carrier matrix. However, cell numbers required for accurate microbial identification are expected to change if different DNA extraction methods are used. For example, phenol-chloroform based DNA extraction is more effective for obtaining DNA from low numbers of cells and may result in more DNA from the same bacterial biomass compared to silica-coated beads or columns [41]. To this regard, other studies which have applied 16S rRNA gene amplicon DNA sequencing to milk were successful with the use of 2 mL to 40 mL milk and approximately $10^3$ to $10^6$ bacterial cells [42–44]. Notably, those methods are not easily automatable and therefore are constrained by the numbers of samples that can be examined simultaneously.

## Collected cells storage conditions

Different BCMC storage conditions and exposure to freeze-thaw cycles did not change the mock community composition. This was also found previously [45–47]. These procedural changes are also expected to have less of an effect than other up-stream sample preparation steps, such as homogenization required for solid or semi-solid samples (e.g. stool) [48]. Although not tested directly it was previously shown that when rapid ($< 24$ h) cold transfer of liquid milk samples between collection site to the lab is not feasible, the milk may be frozen directly [16] or with the addition of cryoprotectant, depending on the resources available.

## Cell lysis method

Lysis methods (mechanical or enzymatic) conferred the largest impact on bacterial representation in the BCMC. Mild cell lysis methods such as bead-beating for 10 sec at 4 m/sec (B5) or vortexing at 1800 rpm for 30 sec (V3) gave the most accurate results with the mock community. Those methods reversed the trend for an underrepresentation of the Gram-negative taxa *Escherichia* and *Pseudomonas* in the BCMCs when more rigorous methods were used, potentially by preventing DNA shearing [49]. The effects of lysis methods on estimates of bacterial distributions were confirmed on milk samples, for which the proportions of taxa were skewed in a similar direction as found for the BCMC. These results are notable considering that rigorous lysis methods are applied pervasively in human and soil microbiome studies [40, 50] and were recommended by purification kit protocols. Our findings also indicate the need for matrix-specific, DNA extraction protocols. For example, the human gut contains large populations of endospore-forming bacteria [51] and harder-to-lyse microorganisms such as *Bifidobacterium*, *Faecalibacterium*, *Butyrivibrio* and *Eubacterium*, whose detection is only possible with rigorous mechanical lysis [49, 52]. Therefore, a rigorous lysis method used for fecal DNA extraction could be excessive when the goal is to lyse bacteria contained in milk. Moreover, the mild cell lysing conditions can be easily performed using a vortex mixer instead of a bead-beater. The use of a vortex mixer can significantly reduce the cost, training and hands-on time required during DNA extraction.

## DNA purification

With the goal of developing a streamlined, high throughput sample processing pipeline, we compared three MagMAX DNA purification kits (Total, Core, Ultra2) that are compatible for automation with the KingFisher Flex magnetic particle processor. We tested the KingFisher Flex system because it is a small, benchtop instrument that is versatile in the sample numbers, cell lysate volumes, and mixing conditions that can be applied [53, 54]. The MagMAX kits are readily automatable with the KingFisher Flex system but have not been validated for food microbiome research pipelines. Here, we demonstrated that, out of the three kits tested, the Total kit resulted in a more accurate representation of the mock communities. Although there reasons for this result cannot be directly known based on descriptions of the reagents and protocols provided, the Total kit protocol requires an additional wash step which may lead to DNA with higher purity and quality.

## Proteinase K and RNase A

The inclusion of Proteinase K was another protocol modification that improved the outcomes of the BCMC analysis when combined with mild lysis method and the Total kit. The benefit of using Proteinase K for the extraction of bacterial gDNA from milk may due to removal of PCR inhibitors such as milk proteinases [55]. Proteolysis may be particularly beneficial for milk due to its high protein content (33.9 g of protein per kg) [56] relative to other common sample types such as human feces (1.8 to 9.9 g/kg) [57] and soil (around 1.5 g/kg) [58].

RNase A treatment, aimed to remove RNA contamination for the genomic DNA, did not improve efforts to identify the BCMC composition by DNA sequencing and instead appeared to introduce bacterial contaminants. Commercially prepared RNase A is reported to be isolated from bovine pancreas [59]. It is possible that DNA from *Ruminococcus*, a common bovine rumen resident [60, 61] contaminated the reagent. Because RNA removal is not as crucial for amplicon-based DNA sequencing, we recommend to not include this step to avoid additional manipulation of samples and to minimize reagent or pipetting contamination. However, it should be noted that the presence of RNA prevents the precise quantification of DNA by spectrophotometric methods [62].

Lastly, it should be noted that while these steps worked well for the analysis of bulk milk from holding and processing vats, other bovine milk sample types may require different methodological approaches for microbial DNA extraction and analysis. In particular, there are certain challenges when using milk collected directly from individual healthy or mastitic cow quarters or teat canals [20, 63]. For those milk samples, it is also important to consider the difficulties for aseptic collection, avoiding teat canal, udder skin and environmental microorganisms [20, 64, 65]. Compared to bulk milk, freshly expressed milk may have higher bovine cell numbers or can be enriched with bacteria which are no longer dominant after storage or processing. Milk collected directly from the teat may also contain compounds that inhibit PCR and therefore result in the need for different DNA extraction and purification steps [19, 20].

In conclusion, our approach of using the BCMC and a systematic evaluation of each step in the sample preparation protocol provides a framework which can be applied to other food (and environmental) samples with the goal of validating methods to measure bacterial composition and diversity. Although microbial detection by CFU enumeration on laboratory culture medium remains the gold standard in assessments of food quality, that technique is generally insufficient for predicting and diagnosing spoilage and defect issues. By simplifying and accelerating microbiota identification methods, we expect that it will be possible to obtain more accurate measures for controlling bacterial contents in dairy products and other foods and beverages.

## Acknowledgments

We would like to thank Dr. Jessie Heidenreich at the Hilmar Cheese Company, Hilmar CA, for her help with milk sample collection. We also thank Rene Suleiman for her help with DNA extraction and Caper Jamin and Yanin Srisengfa for their help with bacterial cell culture. We acknowledge NIZO food research for providing strain *Lactococcus lactis* IL1403 and Dr. Steven Lindow at University of California, Berkeley for providing *Pseudomonas fluorescens* A506

## Author Contributions

**Conceptualization:** Maria L. Marco.

**Data curation:** Zhengyao Xue.

**Formal analysis:** Zhengyao Xue.

**Investigation:** Zhengyao Xue, Maria L. Marco.

**Methodology:** Zhengyao Xue, Maria L. Marco.

**Project administration:** Maria L. Marco.

**Resources:** Maria L. Marco.

**Supervision:** Maria L. Marco.

**Validation:** Maria L. Marco.

**Visualization:** Zhengyao Xue.

**Writing – original draft:** Zhengyao Xue, Maria L. Marco.

**Writing – review & editing:** Zhengyao Xue, Maria L. Marco.

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
