## [Decision Letter · Decision Letter 0]

29 Jun 2022

PONE-D-22-11184Improved assessments of milk microbiota composition via sample preparation and DNA extraction methodsPLOS ONE

Dear Dr. Marco,

Thank you for submitting your manuscript to PLOS ONE. After careful consideration, we feel that it has merit but does not fully meet PLOS ONE’s publication criteria as it currently stands. Therefore, we invite you to submit a revised version of the manuscript that addresses the points raised during the review process.

We look forward to receiving your revised manuscript.

Kind regards,

Mary Anne Amalaradjou

Academic Editor

PLOS ONE

Journal Requirements:

“The funding agency did not participate in study design, data collection, or interpretation of the data.”

Additional Editor Comments:

Improved assessments of milk microbiota composition via sample preparation and DNA extraction methods

Overall the article is well written with clear presentation of results and discussion. As outlined by the reviewers expansion of details around sample type, revision of fig.1 and including information on the potential relevance of this study relative to other published literature will help strengthen the submission.

Reviewers' comments:

Reviewer's Responses to Questions

**Comments to the Author**

1. Is the manuscript technically sound, and do the data support the conclusions?

Reviewer #1: Yes

Reviewer #2: Yes

2. Has the statistical analysis been performed appropriately and rigorously? 

Reviewer #1: Yes

Reviewer #2: Yes

3. Have the authors made all data underlying the findings in their manuscript fully available?

Reviewer #1: Yes

Reviewer #2: Yes

4. Is the manuscript presented in an intelligible fashion and written in standard English?

Reviewer #1: Yes

Reviewer #2: Yes

5. Review Comments to the Author

Reviewer #1: Improved assessments of milk microbiota composition via sample preparation and DNA extraction methods

General comments.

While there have been many studies reporting the composition of the “bovine milk microbiota” results have varied enormously partly due to a lack of standardized methods and this type of work is urgently needed. It is important to distinguish sample types – studies using samples collected from individual cow mammary glands need to be evaluated differently than samples collected from comingled, refrigerated bulk milk or cheese vats.

Specific Comments

Title – This paper addresses results of studies that have evaluated unprocessed bulk milk microbiota (not individual quarter or composite milk samples from cows). While the concepts described in this paper likely apply to any bovine milk sample, it would help clarify the purpose of this paper to modify the title to read “Improved assessments of “unprocessed bovine bulk milk microbiota…”

Introduction –

Line 50 –59: This paragraph refers to variation in sample type and should be expanded to focus on variation in the types of bovine milk that has been evaluated. The bovine milk microbiota has been evaluated on a variety of samples including bulk milk from single farms, comingled milk from processing vats, individual cow quarter milk samples, teat canal samples, and composite milk samples from cows. Variation in results is at least in part due to variation in the expected viable communities in these diverse samples. For example, it would be good to add a few comments about the challenges in aseptic collection of bovine milk samples (contamination from teat canal bacteria, sampling process and environment), and how various studies evaluating the bovine milk microbiota have used differing methods to collect milk samples. It is also important to note that the studies cited (references 16-18) all referred to raw BULK milk samples (not individual quarter milk samples from cows). Raw bulk milk contains bacteria that are acquired during the harvesting and storage process, thus results of studies describing the bovine milk microbiota that used bulk milk samples need to be differentiated from results of studies that use individual milk samples.

Materials and Methods

L94 – why was 2% fat milk used? That is lower than that expected in unprocessed milk. Would you expect the lower fat concentration to influence your results?

L109 –“performed” not “preformed”

L110 – delete “sample”

Results.

L212-213 – please add “good quality, raw “BULK” milk

L215-218 – this is very good information that will be useful to future researchers

Discussion - Please add a sentence or 2 to guide readers relative to applicablity of these results to other studies - especially those using quarter or composite milk samples collected directly from bovine mammary glands.

L304-308 – nice summary statement

Reviewer #2: I am writing to express my opinion regarding the study entitled “Improved assessment of milk microbiota composition via sample preparation and DNA extraction methods” from Marco and Xue.

In general, the article is well structured, with a very good explanation of the results and a well-defined discussion of the same. The study regards the assessment of a good standardized methodology for milk sampling collection in order to obtain high quality-data for microbiome composition studies. The authors tried to settle the best conditions for sampling, storing, and analyzing milk samples to determine a microbial composition without any biases due to the excess presence of certain bacteria compared to others.

In my opinion, the study deserves to be published, because it paves the basis of best practices when it comes to milk microbiome studies. There are just a few minor explanations and corrections that authors should make before the article can go for publication.

Materials and Methods section

In this section, it is not clear why they prepared the mock community and then they inoculated the BCMC2 in milk samples. In general, the whole preparation of the study needs to be revised and clarified.

Paragraph 2.4: for example, to me, it was not clear why there was no milk in the sample storage condition comparisons, but just because the authors mentioned the inoculation of the milk in the previous paragraph.

Fig. 1: this figure is intended to explain in an easier way the Materials and Methods of the study, but in my opinion, needs a little revision, to help the reader what has been done.

Line 159: Could you please explain why you used both qiime 1 and qiime 2? Did the results refer to which qiime? Moreover, you used the 2018.4 version of qiime 2, which is from 4 years ago. Could you try to explain why you choose to use an older version instead of a new one?

Line 168: this is just a curiosity. Why did you use Greengenes instead of Silva of other databases?

Results and Discussion sections

Those two sections are very well written and explained. Everything done for this study becomes very clear when I read the results and the following discussion of the same. I have just a minor comment.

Lines 195-198: in my opinion, this little part could be moved to the Materials and Methods section, because it makes it very easy to understand why you inoculated the milk and for what purpose.

6. PLOS authors have the option to publish the peer review history of their article (what does this mean?). If published, this will include your full peer review and any attached files.

Reviewer #1: No

Reviewer #2: No

---

## [Author Response · Author response to Decision Letter 0]

13 Aug 2022

Journal Requirements:

Response: We revised the paper to meet expectations for the PlosOne formatting style

Response: The Acknowledgements section was revised to include this information.

“The funding agency did not participate in study design, data collection, or interpretation of the data.” We note that you have provided additional information within the Acknowledgements Section that is not currently declared in your Funding Statement. Please note that funding information should not appear in the Acknowledgments section or other areas of your manuscript. We will only publish funding information present in the Funding Statement section of the online submission form. Please remove any funding-related text from the manuscript and let us know how you would like to update your Funding Statement. Currently, your Funding Statement reads as follows:

Response: We revised the Acknowledgements section and provide the funder information in the cover letter.

Response: We revised the reference list to include additional citations relevant to the reviewer’s comments.

Additional Editor Comments: 

Improved assessments of milk microbiota composition via sample preparation and DNA extraction methods

Overall the article is well written with clear presentation of results and discussion. As outlined by the reviewers expansion of details around sample type, revision of fig.1 and including information on the potential relevance of this study relative to other published literature will help strengthen the submission.

Response: We thank the editor for the kind works about our paper. We have revised the manuscript to address the concerns raised by the reviewers. 

Reviewers' comments:

Reviewer's Responses to Questions

Comments to the Author

1. Is the manuscript technically sound, and do the data support the conclusions?

Reviewer #1: Yes

Reviewer #2: Yes

2. Has the statistical analysis been performed appropriately and rigorously? 

Reviewer #1: Yes

Reviewer #2: Yes

3. Have the authors made all data underlying the findings in their manuscript fully available?

Reviewer #1: Yes

Reviewer #2: Yes

4. Is the manuscript presented in an intelligible fashion and written in standard English?

Reviewer #1: Yes

Reviewer #2: Yes

5. Review Comments to the Author

Reviewer #1: Improved assessments of milk microbiota composition via sample preparation and DNA extraction methods

General comments.

While there have been many studies reporting the composition of the “bovine milk microbiota” results have varied enormously partly due to a lack of standardized methods and this type of work is urgently needed. It is important to distinguish sample types – studies using samples collected from individual cow mammary glands need to be evaluated differently than samples collected from comingled, refrigerated bulk milk or cheese vats.

Response: We thank the reviewer for the encouraging works and recognition of our work.

Specific Comments

Title – This paper addresses results of studies that have evaluated unprocessed bulk milk microbiota (not individual quarter or composite milk samples from cows). While the concepts described in this paper likely apply to any bovine milk sample, it would help clarify the purpose of this paper to modify the title to read “Improved assessments of “unprocessed bovine bulk milk microbiota…”

Response: We revised the title to clarify that bulk milk was the focus of the work.

Introduction –

Line 50 –59: This paragraph refers to variation in sample type and should be expanded to focus on variation in the types of bovine milk that has been evaluated. The bovine milk microbiota has been evaluated on a variety of samples including bulk milk from single farms, comingled milk from processing vats, individual cow quarter milk samples, teat canal samples, and composite milk samples from cows. Variation in results is at least in part due to variation in the expected viable communities in these diverse samples. For example, it would be good to add a few comments about the challenges in aseptic collection of bovine milk samples (contamination from teat canal bacteria, sampling process and environment), and how various studies evaluating the bovine milk microbiota have used differing methods to collect milk samples. It is also important to note that the studies cited (references 16-18) all referred to raw BULK milk samples (not individual quarter milk samples from cows). Raw bulk milk contains bacteria that are acquired during the harvesting and storage process, thus results of studies describing the bovine milk microbiota that used bulk milk samples need to be differentiated from results of studies that use individual milk samples.

Response: We thank the reviewer for this insight and reminding us of this important point. We revised the Introduction and Discussion to explain the challenges associated with microbial analysis for different milk sample (collection) types. 

Materials and Methods

L94 – why was 2% fat milk used? That is lower than that expected in unprocessed milk. Would you expect the lower fat concentration to influence your results?

Response: We used the UHT milk to minimize contamination of the BCMC and thus skewing of the data for the DNA extraction and purification tests. 

L109 –“performed” not “preformed”

Response: Corrected.

L110 – delete “sample”

Response: Deleted

Results.

L212-213 – please add “good quality, raw “BULK” milk

Response: “bulk’ has been added

L215-218 – this is very good information that will be useful to future researchers

Response: Thank you!

Discussion - Please add a sentence or 2 to guide readers relative to applicablity of these results to other studies - especially those using quarter or composite milk samples collected directly from bovine mammary glands.

Response: The Discussion has been revised to address the need to test the protocols for freshly expressed quarter/composite milk samples and several relevant recent publications on this topic have been included. We agree, also based on our own recent experience, that milk collected directly from bovine mammary glands has additional challenges for PCR amplification.

L304-308 – nice summary statement

Response: Thank you!

Reviewer #2: I am writing to express my opinion regarding the study entitled “Improved assessment of milk microbiota composition via sample preparation and DNA extraction methods” from Marco and Xue.

In general, the article is well structured, with a very good explanation of the results and a well-defined discussion of the same. The study regards the assessment of a good standardized methodology for milk sampling collection in order to obtain high quality-data for microbiome composition studies. The authors tried to settle the best conditions for sampling, storing, and analyzing milk samples to determine a microbial composition without any biases due to the excess presence of certain bacteria compared to others.

In my opinion, the study deserves to be published, because it paves the basis of best practices when it comes to milk microbiome studies. There are just a few minor explanations and corrections that authors should make before the article can go for publication.

Response: Thank you for the positive remarks and support of our paper.

Materials and Methods section

In this section, it is not clear why they prepared the mock community and then they inoculated the BCMC2 in milk samples. In general, the whole preparation of the study needs to be revised and clarified.

Response: We revised the materials and methods to explain the rationale to prepare the BCMC prior to inoculation into the milk. We also revised the order of the materials and methods section and tried to clarify further with revisions to Fig 1.

Paragraph 2.4: for example, to me, it was not clear why there was no milk in the sample storage condition comparisons, but just b authors mentioned the inoculation of the milk in the previous paragraph.

Response: Thank you for pointing this out. Because we collect the bacterial cells from milk samples prior to freezing, we did not include milk in the sample storage condition tests. However, we recognize that this may be regarded to be a limitation of the study and have revised the Discussion section to explain this constraint within the study design.

Fig. 1: this figure is intended to explain in an easier way the Materials and Methods of the study, but in my opinion, needs a little revision, to help the reader what has been done.

Response: Thank you for this suggestion. Fig 1 was revised to remove the “2%”, the text was changed to “Sample storage of bacteria collected by centrifugation” to indicate that these steps are referring to bacterial pellets, not milk. Lastly, we added a step under the milk volume to indicate bacteria collection by centrifugation

Line 159: Could you please explain why you used both qiime 1 and qiime 2? Did the results refer to which qiime? Moreover, you used the 2018.4 version of qiime 2, which is from 4 years ago. Could you try to explain why you choose to use an older version instead of a new one?

Response: QIIME 1.9.1 was only used to extract custom barcode sequences using the extract_barcodes.py script with no barcode error allowed. This was because at the time of the analysis, that function was not available in QIIME 2. The analysis parameters in the QIIME 2 2018.4 version was used based on a previous publication that compared a few different bioinformatics workflow for 16S rRNA gene bacterial community analysis (https://journals.asm.org/doi/full/10.1128/mSphere.00410-18). We expect the results and conclusions to stay the same even if a newer version of QIIME 2 is used. 

Line 168: this is just a curiosity. Why did you use Greengenes instead of Silva of other databases?

Response: We used Greengenes because our prior publication showed it was more accurate than RDP (https://journals.asm.org/doi/full/10.1128/mSphere.00410-18). We now explain this in the Materials and Methods. We agree that Silva is a good and frequently used alternative, but for the purposes of continuity, we used Greengenes in this follow-up paper.

Results and Discussion sections

Those two sections are very well written and explained. Everything done for this study becomes very clear when I read the results and the following discussion of the same. I have just a minor comment.

Lines 195-198: in my opinion, this little part could be moved to the Materials and Methods section, because it makes it very easy to understand why you inoculated the milk and for what purpose.

Response: Thank you for the suggestion. We moved that sentence to the Materials and Methods section.

6. PLOS authors have the option to publish the peer review history of their article (what does this mean?). If published, this will include your full peer review and any attached files.

Do you want your identity to be public for this peer review? For information about this choice, including consent withdrawal, please see our Privacy Policy.

Reviewer #1: No

Reviewer #2: No

---

## [Editor Report · Decision Letter 1]

25 Aug 2022

Improved assessments of bulk milk microbiota composition via sample preparation and DNA extraction methods

PONE-D-22-11184R1

Dear Dr. Marco,

We’re pleased to inform you that your manuscript has been judged scientifically suitable for publication and will be formally accepted for publication once it meets all outstanding technical requirements.

Kind regards,

Mary Anne Amalaradjou

Academic Editor

PLOS ONE
---

## [Editor Report · Acceptance letter]

6 Sep 2022

PONE-D-22-11184R1 

Improved assessments of bulk milk microbiota composition via sample preparation and DNA extraction methods 

Dear Dr. Marco:

I'm pleased to inform you that your manuscript has been deemed suitable for publication in PLOS ONE. Congratulations! Your manuscript is now with our production department. 

Kind regards, 

on behalf of

Dr. Mary Anne Amalaradjou 

Academic Editor

PLOS ONE